# Silence of Hippo Pathway Associates with Pro-Tumoral Immunosuppression: Potential Therapeutic Target of Glioblastomas

**DOI:** 10.3390/cells9081761

**Published:** 2020-07-23

**Authors:** Eui Hyun Kim, Bo Hwa Sohn, Young-Gyu Eun, Dong Jin Lee, Sun Young Yim, Seok-Gu Kang, Ju-Seog Lee

**Affiliations:** 1Department of Systems Biology, Division of Cancer Medicine, Unit 950, The University of Texas MD Anderson Cancer Center, 1515 Holcombe Blvd, Houston, TX 77030, USA; euihyunkim@yuhs.ac (E.H.K.); bsohn@mdanderson.org (B.H.S.); ygeun@hanmail.net (Y.-G.E.); djlee@hallym.or.kr (D.J.L.); eug203@naver.com (S.Y.Y.); 2Department of Neurosurgery, Severance Hospital, Brain Tumor Center, Yonsei University College of Medicine, Seoul 03722, Korea; SEOKGU9@yuhs.ac; 3Department of Otolaryngology—Head and Neck Surgery, School of Medicine, Kyung Hee University, Seoul 02447, Korea; 4Department of Otolaryngology—Head and Neck Surgery, School of Medicine, Hallym University, Seoul 24252, Korea; 5Division of Gastroenterology and Hepatology, Department of Internal Medicine, Korea University College of Medicine, Seoul 136-701, Korea

**Keywords:** glioblastoma, Hippo pathway, immune checkpoint, immune signature, macrophage, M2 polarization

## Abstract

The critical role of the Hippo pathway has been recently investigated in various cancers, but little is known about its role in glioblastoma (GBM). In order to evaluate the clinical relevance of the Hippo pathway in GBM, we generated a core gene expression signature from four different previously-established silence of Hippo pathway (SOH) signatures. Based on a newly generated core SOH signature, a SOH and active Hippo pathway (AH) was predicted in GBM samples from The Cancer Genome Atlas (TCGA) and validated in a separate cohort. A comparative analysis was performed on multi-panel genomic datasets from TCGA and the possible association of SOH with immune activity and epithelial mesenchymal transition was also evaluated. The SOH signature was associated with poor prognosis in GBM in both cohorts. Expression levels of *CTGF* and *CYR61*, the most reliable and well-known downstream targets of *YAP1*, were markedly increased in the SOH subgroup of GBM patients. SOH signature was strongly associated with a high immune signature score and mesenchymal features. Genes differentially expressed between SOH and AH groups revealed many markers for inhibitory immune checkpoints and M2-polarized macrophages were upregulated in the SOH subgroup, suggesting that SOH may induce the resistance of cancer cells to host immune response in GBM. In summary, SOH is significantly associated with the poor prognosis of GBM patients and is possibly mediated by pro-tumoral immunosuppression.

## 1. Introduction

Glioblastoma (GBM) is the most deadly cancer that develops in the central nervous system. Since 2005, concomitant and maintenance treatment with temozolomide and radiotherapy has been the standard treatment for patients with newly diagnosed GBM [1]. Most chemotherapeutic and targeted agents have failed to demonstrate superiority to this regimen, mainly because of tumor heterogeneity and the blood–brain barrier [2].

The Hippo pathway is known to play a critical role in the regulation of tissue homeostasis, organ size control, and stem cell renewal [3,4]. Moreover, an increasing number of reports have suggested that the Hippo pathway contributes to cancer development and progression [3,5]. The kinase complex, including *STK3*/*4* (), *SAV1*, and *LATS1*/*2*, is the core functional component of the pathway, inhibiting the oncogenic transcription activators *YAP1* and *TAZ* (also known as *WWTR1*) through phosphorylation. When YAP1 becomes unphosphrorylated, it enters a nucleus and activates various transcriptional factors such as *TEAD* and *SMAD,* increasing gene expression involved in cell proliferation and survival [6]. *CYR61*, *CTGF* and *ANKRD1* are the most well-known downstream targets of *YAP1* [6,7]. Despite numerous observations that the Hippo pathway is involved in various cancers [8], its role in the development and progression of GBM has not been properly addressed, and the clinical significance of the inactivation of the Hippo pathway has not been fully evaluated. Although the inactivation of the Hippo pathway is generally associated with mutations in the Hippo pathway components and the copy number amplification of *YAP1*, these are extremely rare in GBM (cBioPortal, http://www.cbioportal.org/).

In our previous studies [9,10,11], we established gene expression signatures reflecting SOH in stomach, liver, ovarian, and colorectal cancers and demonstrated that SOH signatures were associated with poor prognosis in these four cancers. In the current study, by integrating four different SOH signatures from different cancers, we identified a core SOH gene expression signature that was common to all four cancer types and applied this core SOH signature to GBM tumors to assess the clinical relevance of the Hippo pathway in GBM.

## 2. Materials and Methods

### 2.1. General Analytic Methods

To generate the core silence of Hippo pathway (SOH) signature, we used four SOH gene expression signatures from our previous studies in which the role of the Hippo pathway was investigated in four different cancers: colorectal, liver, ovarian, and stomach cancers [9,10,11]. Gene expression data for colorectal, liver, ovarian, stomach cancers, and glioblastoma (GBM) from The Cancer Genome Atlas (TCGA) project were obtained from a data portal site (https://tcga-data.nci.nih.gov). For validation, expression data from another GBM dataset (GSE16011) were used (http://www.ncbi.nlm.nih.gov/geo). Molecular characteristics of GBM tumors were retrieved from previous TCGA publications [12]. BRB-ArrayTools (http://linus.nci.nih.gov/BRB-ArrayTools.html) was used to analyze gene expression data [13]. A heatmap was generated using Cluster and TreeView [14]. Other statistical analyses were performed in the R language (http://www.r-project.org). We used the Bayesian covariate compound predictor (BCCP) algorithm, which was described previously [15]. Survival analysis was performed using the Kaplan–Meier method to compare groups. *p* values less than 0.05 were considered statistically significant unless otherwise indicated.

### 2.2. Generation of a Core SOH Signature and Validation in GBM Datasets

A three-step approach was applied to generate the core SOH signature (Figure 1).

First, previously established SOH gene expression signatures from four cancer types were applied to corresponding RNA-seq data from TCGA using the Bayesian covariate compound predictor (BCCP) algorithm, as described in previous studies [16,17,18,19]. Tumor sample data from TCGA comprised 262 stomach tumors, 193 liver tumors, 266 ovarian tumors, and 286 colorectal tumors. Gene expression data from the original signatures and TCGA were re-normalized by standardization before the prediction model was applied. Tumors were then stratified into SOH or active Hippo pathway (AH) subgroups according to corresponding SOH signatures. Second, genes differentially expressed between the SOH and AH subgroups in each TCGA dataset were selected using the Student *t*-test with a cutoff of *p* < 0.001 and at least a two-fold difference. This yielded 5585 genes for stomach cancer, 5988 genes for liver cancer, 4127 genes for ovarian cancer, and 5859 genes for colorectal cancer. We then selected the 689 genes that were common to all four gene lists for the core SOH signature.

The core SOH signature generated from the four cancer types was used as a training set for further analysis to assess the clinical relevance of the SOH signature in GBM. Briefly, gene expression data in this training set were combined to form a classifier according to the BCCP algorithm. The robustness of the classifier was estimated using a misclassification rate determined during leave-one-out cross-validation in the training set. Then, the new BCCP classifier (689 genes in 1007 samples) estimated the likelihood that individual samples from GBM expression data have either the SOH or AH signature, according to a Bayesian probability with a 0.5 cutoff. After stratification of tumors, the prognostic significance was estimated using Kaplan–Meier analysis. The same prediction was performed for 172 samples from TCGA including 154 primary and 13 recurrent GBMs and five samples of normal brain tissue.

### 2.3. Genomic Analysis of TCGA Cohort Data

Among 154 primary GBM samples with gene expression data (Illumina HiSeq 2000 RNA Sequencing platform) retrieved from TCGA, copy number variation data were available in 148 samples and somatic mutation data were available in 147 samples. GISTIC 2.0 copy number variation data were used to generate a heatmap and graph showing the difference between the SOH and AH subgroups [20]. A previous TCGA study identified 71 significantly mutated genes in GBM, selected with a mutation frequency above background with a q-value of <0.1 [12]. After excluding five genes in which no mutation was found in any of the 154 samples, we compared the number of mutations per sample between the two subgroups using the chi-square test.

### 2.4. Proteomics of TCGA Cohort Data

Reverse phase protein array (RPPA) data of TCGA samples were obtained from the TCGA RPPA core at The University of Texas MD Anderson Cancer Center (Houston, TX, USA). Among the 154 primary GBM samples used in our study, RPPA data were available in 72 samples. The differences in protein expression levels between the two subgroups were evaluated using the Student *t*-test and *p* < 0.01 was considered statistically significant.

### 2.5. Comparison of mRNA Expression Levels and Pathway Analysis

The mRNA expression levels of Hippo pathway-related genes and Hippo target genes *YAP1* and *TAZ* were compared between the SOH and AH subgroups in two cohorts: TCGA and GSE16011. The expression level of *CTGF*, *CYR61*, and *ANKRD1*, the most reliable and well-known downstream targets of *YAP1* were also compared between two subgroups [7]. Markers for macrophages and their polarized subtypes were selected on the basis of findings from previous studies, and differential gene expression of these markers between the SOH and AH subgroups was investigated [21,22]. For comparative analysis of differential expression of inhibitory immune checkpoints between the SOH and AH subgroups in TCGA and GSE16011, we retrieved the list of inhibitory immune checkpoint molecules from a previous study [23].

For pathway analysis, differentially expressed genes were selected by comparing mRNA expression levels of SOH and AH in TCGA samples using the Student *t*-test (*p* < 0.001) and at least a two-fold difference. Data were analyzed using QIAGEN’s Ingenuity Pathway Analysis (IPA) (QIAGEN, Redwood City, CA, USA; www.qiagen.com/ingenuity).

### 2.6. Immunohistochemistry on GBM Tissue

We retrieved the glioma dataset from The Human Protein Atlas (https://www.proteinatlas.org/) and compared immunohistochemical (IHC) stains for YAP1 and PD-1 in 7 matched GBM samples. Three GBM samples with positive IHC stain for nuclear YAP1 were considered as SOH and compared with other 4 AH GBM samples. The correlation between YAP1 and PD-1 were evaluated by χ^2^ test. 

### 2.7. Immune Signature Score (ISS) and Leukocyte Subset Analysis

The algorithm to generate the ISS was described in our previous study [24]. Briefly, a previously identified 105-gene immune signature was used to generate ISS in GBM samples in the GSE16011 and TCGA datasets. Patients were stratified into high-ISS or low-ISS groups with an ISS cutoff of 0.5. To assess the activity of immune cells in GBM tissues, we carried out a meta-analysis with CIBERSORT (https://cibersort.stanford.edu), which can estimate the proportion of immune cells in a tumor mass from mRNA expression data [25]. Analyses were performed with 100 permutations using default statistical parameters.

### 2.8. Transforming Growth Factor Beta (TGF-β) Signature

The TGF-β signature was obtained from a previous study in which GBM tumorspheres were treated with galunisertib (LY2157299), a well-known TGF-β receptor 1 kinase inhibitor (GSE23935) [26]. A total of 284 differentially expressed genes was selected (*p* < 0.001) and used to construct a classifier. One hundred fifty-four GBM samples were later dichotomized into active TGF-β and inactive TGF-β subgroups, with a 0.5 cutoff. Associations between the TGF-β signature and the core SOH signature and previously established molecular subtypes were evaluated.

### 2.9. Epithelial–Mesenchymal Transition Signature

In order to classify 154 GMB samples into high and low epithelial–mesenchymal transition (EMT)-associated groups, 64-gene list was obtained from previous study on *SNAI2*-induced EMT [27]. Briefly, the average expression level of these 64 genes, which is referred as EMT metagene score in our study, was calculated in each sample of TCGA cohort. According to EMT metagene score, the samples were aligned and its association with Hippo signature and previously established molecular subtypes was evaluated.

## 3. Results

### 3.1. Clinical Significance of Inactivation of the Hippo Pathway in GBM

Previous studies showed that the activity of the Hippo pathway is best reflected in transcriptome, as final effectors of the Hippo pathway are transcription factors such as *YAP1* and *TAZ* [9,10,11]. For a better understanding of the underlying biology related to inactivation of the Hippo pathway (or activation of *YAP1*/*TAZ*) in cancer, we generated a core SOH gene expression signature by re-analyzing previously established SOH signatures from stomach, liver, ovarian, and colorectal cancers, as described in Materials and Methods (Figure 1). Interestingly, the vast majority of genes in the core SOH signature were upregulated in the SOH subgroups across the four cancer types, suggesting that many of these genes might be direct targets of *YAP1*/*TAZ*, which are best known as transcription activators. In fact, the best known direct downstream targets of *YAP1*/*TAZ*, such as *CTGF* and *CYR61* [7], were significantly upregulated in the SOH subgroups.

Once we had a core SOH signature that accurately reflected the inactivation of the Hippo pathway regardless of cancer type or organ site, we applied the signature to gene expression data from GBM tumor samples in TCGA to stratify patients, using the previously established BCCP algorithm [16,17,18,19]. When dichotomized into SOH (53 patients) and AH (101 patients) subgroups (Figure 2A), patients in the SOH subgroup had significantly shorter overall survival (*p* = 0.029; Figure 2B), suggesting that activation of *YAP1*/*TAZ* is associated with poor prognosis in GBM.

Among the four previously recognized molecular subtypes (classical, mesenchymal, neural, and proneural) [28], the mesenchymal subtype was associated with the SOH signature (*p* < 0.001 by chi-square test, Figure 2A), suggesting that the acquisition of the mesenchymal phenotype might be mediated by activation of *YAP1*/*TAZ* in GBM. Consistent with the association of the SOH signature with poor prognosis, tumors in the SOH subgroup lacked the glioma CpG island methylation phenotype that is typically associated with improved prognosis in GBM (Figure 2A) [29]. The incidence of promoter methylation of methylguanine methyl transferase was not associated with either the SOH or AH signature (Figure 2A). The core SOH signature-based prediction of 172 TCGA samples including primary, recurrent GBM and normal brain, demonstrated recurrent samples were more common in the SOH, whereas all normal brain tissue samples were classified as AH (Appendix A).

We further tested the association of the SOH signature with poor prognosis in an independent GBM cohort, GSE16011. When 96 GBM patients were stratified into the SOH or AH subgroups (Figure 2C), Kaplan–Meier plots showed a significant association between the gene signature and overall survival (*p* = 0.027, Figure 2D), strongly supporting the robustness of the signature and association of *YAP1*/*TAZ* activation with poor prognosis in GBM.

### 3.2. Genomic and Proteomic Characteristics Associated with the SOH Signature

In TCGA, somatic mutation analysis revealed that overall somatic mutation burden was not significantly different between the SOH and AH subgroups or among the four previously recognized molecular subtypes (Appendix A). Copy number variations were not significantly different between the SOH and AH subgroups either (Appendix A). In the analysis on proteomic characteristics of GBM by using RPPA data, we identified 48 protein features associated with the SOH signature (*p* < 0.01; Appendix A).

### 3.3. Molecular Characteristics Associated with the SOH Signature

We next identified genes whose expression was significantly different between the SOH and AH subgroups (*p* < 0.001) in TCGA, yielding 904 genes (Appendix A). When the gene expression levels of upstream members of Hippo pathway such as *LATS1*, *LATS2*, *SAV1* were compared between SOH and AH subgroups, only *MST2* and *LATS2* were upregulated in SOH subgroup, which revealed that they may be not the leading cause of YAP1 activation (Figure 3). Expression of *YAP1* was not significantly different between the two subgroups, which may be attributed to the fact that regulation of YAP1 function largely depends on its phosphorylation state and on the site of the phosphorylation. On the contrary, expression of *TAZ* was significantly higher in the SOH subgroup, suggesting that *TAZ* may play more of a role in GBM (Figure 3). Likewise, among four TEAD family members, expression of *TEAD3* and *TEAD4* was significantly higher in the SOH subgroup, further supporting isoform-specific activation of the *YAP1*/*TAZ*-TEAD pathway in GBM. Consistent with the increased expression of *TAZ* and *TEAD3/4*, expression of *CTGF* and *CYR61*, the best known downstream targets of *YAP1*/*TAZ* [7], were significantly higher in the SOH subgroup than in the AH subgroup. This observation is further supported by our finding that expression of *CTGF* and *CYR61* was significantly higher in the SOH subgroup than in the AH subgroup in an independent cohort of GBM patients (GSE16011; Appendix A).

To assess the underlying biology associated with the SOH subgroup, we carried out gene network analysis using IPA with 904 differentially expressed genes. Interestingly, many signaling pathways related to immune response, such as TREM1 signaling, dendritic cell maturation, iCOS-iCOSL signaling in T helper cells, and CD28 signaling in T helper cells, were significantly enriched in the SOH subgroup (Figure 4). 

Consistent with these findings, cytokines such as *TNF*, *TGFB1*, *INFG*, *IL*-*1B*, and *IL*-*4* were predicted as potential upstream regulators in gene network analysis (Appendix A). More interestingly, many inhibitory immune checkpoint genes, such as *CTLA4*, *PD*-*1*, and *PD*-*L2*, were upregulated in the SOH subgroup, suggesting that *YAP1*/*TAZ* may induce resistance of cancer cells to host immune response in GBM (Figure 5). Because transcriptionally active YAP1 is typically localized in nuclei [6], we examined the IHC staining of YAP1 and PD-1 in GBM tissues. In good agreement with observation from genomic data, nuclear YAP1 staining is highly correlated with PD-1 staining (Appendix A).

### 3.4. Immunologic Features Associated with Activation of YAP1/TAZ in GBM

Because gene network analysis strongly suggested that the SOH signature may be associated with immune-related pathways, we further analyzed genomic data to determine whether *YAP1*/*TAZ* plays a role in regulation of the host immune response to cancer cells. We previously developed ISS by adopting 105 immune signature genes selected by comparing expression data between responders and non-responders to immunotherapy [24]. We applied the ISS algorithm to gene expression data from 154 GBM tissues and dichotomized the samples into high- and low-ISS subgroups (cutoff of 0.5). Consistent with gene network analysis, the SOH subgroup was associated with high ISS (*p* < 0.001, chi-square test; Figure 6A), further supporting the potential role of *YAP1*/*TAZ* in immune response to GBM. The mesenchymal subtype had the highest ISS and the proneural subtype had the lowest ISS (Figure 6B), suggesting that SOH is associated with the mesenchymal subtype. This association was further validated in an independent cohort of GBM patients (Appendix A).

To assess immune cell components that are most likely accountable for the high immune activity in the SOH subgroup, we applied meta-analysis to gene expression data from GBM tissue using CIBERSORT. This analysis revealed that the M2 macrophage was the most abundant immune cell component in GBM (Figure 6A), and the SOH subgroup showed much higher proportions of M2 macrophages (42.8% compared with 34.6% for AH; *p* < 0.001). We next compared the expression levels of macrophage surface makers, including general markers for macrophages and specific markers for M2 polarized macrophages. Expression of M2 polarized macrophage markers was significantly higher in the SOH subgroup than in the AH subgroup (Appendix A). Differential expression of M2 markers between the two subgroups was further validated in the GSE16011 GBM dataset (Appendix A).

### 3.5. TGF-β Signature and Inactivation of the Hippo Pathway

Because TGF-β is the most well-known immunosuppressive molecule particularly associated with the mesenchymal phenotype [30], we investigated whether the TGF-β pathway is associated with immune activity as well as activation of *YAP1*/*TAZ* in GBM. When a 284-gene TGF-β classifier was applied to TCGA data, the SOH subgroup had significantly higher TGF-β activity than the AH subgroup (*p* < 0.001, chi-square test; Appendix A). GBM samples with the active TGF-β signature also apparently showed much higher ISSs (*p* < 0.001, Appendix A). The gene expression level of TGF-β was positively correlated with all M2 markers, including *MRC1*, *CD163*, *TREM2*, *IL*-*10*, and *ARG1*, and *CLEC7A* was the marker most well correlated with TGF-β (Appendix A).

### 3.6. Epithelial–Mesenchymal Transition and Inactivation of the Hippo Pathway

When TCGA GBM samples were predicted based on 64 genes of EMT signature, GBM samples with SOH signature demonstrated higher EMT metagene scores (Appendix A). Between BCCP value of SOH and EMT metagene score, there was a strong positive correlation (r = 0.830).

## 4. Discussion

The inactivation of the Hippo pathway in cancers is mediated by various genetic and epigenetic events. The Hippo pathway is a very complex regulatory network as reflected by many upstream regulators in the pathway. However, the vast majority of pathway activity is largely mediated by two transcription regulators *YAP1* and *TAZ*. Many previous studies showed that their activity is best reflected in the transcriptome as they are strong transcription regulators [3,6]. We used transcription signatures adopted from multiple datasets reflecting the inactivation of *STK3*/*4* and SAV1 (key negative regulators of *YAP1* and *TAZ*) to identify tumors with inactivated Hippo pathways regardless of their genetic or epigenetic alterations in the Hippo pathway. In fact, their key transcription targets such as *CTGF*, *CYR61*, and *ANKRD1* are significantly more expressed in SOH, strongly supporting that the core SOH signature is a good indicator of Hippo pathway activity.

When we stratified GBM patients based on the core SOH signature, the activation of *YAP1*/*TAZ* was associated with poor prognosis in GBM. The association was tested and validated in two independent cohorts of patients. A further in-depth analysis of genomic data revealed that activation of *YAP1*/*TAZ* might be associated with increased host immune activity. Many inhibitory immune checkpoint genes were upregulated in the SOH subgroup (Figure 5), which possibly results in pro-tumoral T-cell exhaustion in GBM [31,32]. More interestingly, high immune activity in the SOH subgroup was associated with an increased fraction of M2 polarized macrophages that can suppress immune cell activity. Thus, our analysis may provide novel biological insight into the poor prognosis associated with the SOH signature in GBM.

The analysis of TCGA data showed that inactivation of the Hippo pathway was more frequently observed in the mesenchymal subtype, suggesting that the inactivation of the Hippo pathway may be associated with EMT, consistent with previous observations [33]. In order to evaluate the association of SOH with EMT, we adopted an EMT metagene scoring system from a previous study based on a glioblastoma cell line model [27]. When the EMT metagene scores of each GBM TCGA sample were measured based on the expression levels of 64 EMT-related genes, most of the 64 genes were upregulated in a SOH subgroup (Appendix A). Additionally, we performed the same prediction for 13 samples of recurrent GBM and five samples of normal brain tissue, together with the 154 samples of primary GBM, from TCGA to evaluate the SOH signature in recurrent GBM. Considering that GBM acquires mesenchymal features at recurrence regardless of the original subtype [28], it is not surprising that most of the recurrent GBM samples we examined exhibited the SOH signature and all normal brain tissues were skewed to the AH subgroup (Appendix A). In the RPPA analysis on the TCGA cohort, we identified 48 proteins associated with the SOH signature (Appendix A). Among them, we identified 48 protein features associated with the SOH signature many proteins known to be associated with poor prognosis in GBM, such as SERPINE1, FN1, CAV1, IGFBP2, HSPA1A, and SYK [34,35,36,37,38], were upregulated in the SOH subgroup, strongly suggesting potential cross-talk between the Hippo pathway and other signaling pathways associated with poor prognosis in GBM. Although the association of these proteins with the Hippo pathway has not been investigated extensively, recent reports suggested that some of these proteins are possibly controlled by the Hippo pathway [39,40]. More interestingly, the expression and activation of *EGFR* was negatively correlated with inactivation of the Hippo pathway, suggesting that the activation of *YAP1* and *TAZ* might be mutually exclusive with activation of the *EGFR* pathway in GBM. The 904 genes differentially expressed between the SOH and AH subgroups, as well as the pathway analysis, strongly suggested that SOH is associated with the activation of many immune molecules and pathways. Most notably, well-known macrophage-associated genes, including *MARCO*, *FPR2*, *MRC1*, *POSTN*, *CSF3*, *CD163*, and *TREM1*, were among the most upregulated genes in the SOH subgroup. Our ISS system, which was designed to estimate the probability of response to immune-based therapy, predicted that GBM samples with the SOH signature had high immune activity. Although high immune activity is generally correlated with high mutational burden (increased neo-antigen presentation) [41], no difference was found in mutational burden between the SOH and AH subgroups, suggesting that GBM with the SOH signature gains high immunogenicity through mutation-independent mechanisms. 

Although tumors with an increased immune signature generally show better prognosis than those with decreased immune signatures, we did not observe a survival benefit from high ISS in our study. The in-depth analysis of genomic data suggested that the high ISS in the SOH subgroup may be associated with the innate immune system, and more specifically, with macrophages. Macrophages are among the most abundantly recruited immune cells in the tumor microenvironment. Two distinct states of polarized activation for macrophages have been defined: the classically activated macrophage phenotype (M1) with an antitumor effect and the alternatively-activated macrophage phenotype (M2) with a pro-tumoral and immunosuppressive effect [42,43]. In most solid cancers, M2 macrophages comprise the majority of tumor-associated macrophages [43], and the proportion of M2 macrophages is higher in GBM than in any other major cancer [44]. We postulated that tumor-associated macrophages may play pro-tumoral roles when the Hippo pathway is inactive, leading to poor prognosis despite the high overall immune activity in the SOH subgroup. Indeed, mRNA expression of all M2-specific markers was upregulated in the SOH subgroup in two independent GBM cohorts. Our observation that inhibitory immune checkpoint genes showed increased expression in the SOH subgroup also supports this hypothesis. Moreover, our hypothesis is in good agreement with previous observations that increased tumor-infiltrating lymphocytes and macrophages are most common in the mesenchymal subtype of GBM, for which survival is the worst among the four subtypes [45]. Recent reports showed that targeting M2 macrophages in a GBM mouse model led to a remarkable survival benefit by inhibiting *CSF*-*1R*, an M2-related marker [46], which also strongly supports our hypothesis.

TGF-β promotes pro-tumoral immunosuppression. As a pleiotropic cytokine, TGF-β maintains immune homeostasis through the regulation of all cell types of the innate and adaptive immune system [47]. In our pathway analysis, TGF-β was suggested as a key upstream regulator. In particular, TGF-β is known to be a strong inducer of the M2 polarization of macrophages [48]. The critical role of TGF-β in the epithelial–mesenchymal transition has also been demonstrated in many cancer models, including GBM [30,49]. On the basis of this supporting evidence, we postulated that the TGF-β signaling pathway may be associated with the inactivation of the Hippo pathway. The SOH signature was strongly correlated with the active TGF-β signature, which was also associated with high ISS and mesenchymal features. Our results suggest that TGF-β might be another potential target that inhibits M2 polarized macrophages in GBM. Currently, there are several TGF-β inhibitors being tested in clinical trials for several cancer types, including GBM, and the preliminary results have been encouraging [50].

Our study has several limitations. Since our SOH signature is not derived from GBM tissues, predicted *YAP1* activity in GBM might be underestimated due to the lack of expression data reflecting neuronal specific *YAP1* activity. Therefore, SOH and its possible crosstalk with the cancer microenvironment should be further investigated directly based on GBM cell lines and mouse models as suggested by Okazaki et al. [51]. Secondly, because this is an association study, which is the main limitation of our study, further investigation is needed to verify our hypothesis and to elucidate the pro-tumoral mechanism of M2 macrophages. Furthermore, it is currently unknown if the activation of M2 macrophages is mediated by *YAP1*/*TAZ*. Although some of the association from our analysis of genomic data is functionally relevant as evidenced by good correlation between active YAP1 and PD-1 staining of infiltrated cells in IHC staining of GBM tissues, more analysis and experimental approaches will be needed for functional validation. Lastly, our signature may be more related to *YAP1*/*TAZ* activity than the inactivation of the Hippo pathway, as some of upstream regulators such as *MST2* and *LATS* are more expressed in SOH. *YAP1* and *TAZ* might be activated independent of upstream regulators.

In summary, GBM showing the SOH signature was associated with an increased ISS, suggesting that these patients might be good potential candidates for immunotherapy. Many inhibitory immune checkpoint genes are upregulated in the SOH subgroup, suggesting that *YAP1*/*TAZ* may induce the resistance of cancer cells to host immune response in GBM. M2 polarized macrophages may be another important component of pro-tumoral immunosuppression in GBM with a possible association with SOH, and thus can be another possible target of immunotherapy. Targeting the Hippo pathway and associated TGF-β pathway might be a useful approach to enhance the clinical benefit of immunotherapy in GBM.

## Figures and Tables

**Figure 1 cells-09-01761-f001:**
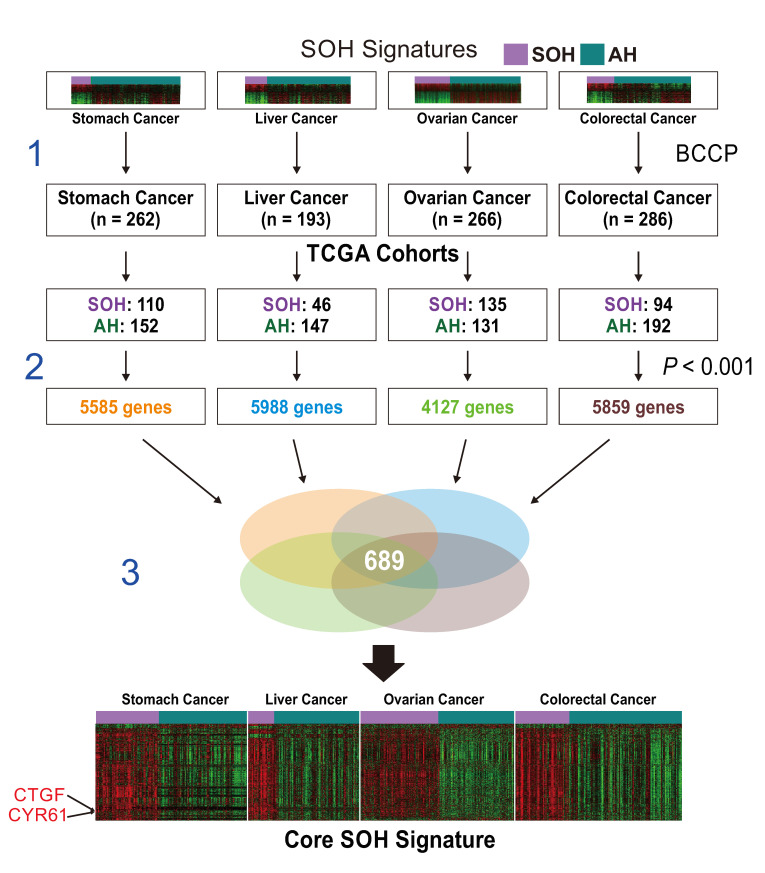
Schematic overview of the workflow for generation of the core silence of Hippo (SOH) gene expression signature. Step 1: original SOH gene expression signatures from each cancer type were applied to corresponding gene expression data from The Cancer Genome Atlas (TCGA) using the Bayesian covariate compound predictor (BCCP) algorithm. Tumors in TCGA were stratified into SOH or AH (active Hippo pathway) subgroups according to the corresponding SOH signature from each cancer type. Step 2: differentially expressed genes between the SOH and AH subgroups in each cancer type were identified by *t*-test (*p* < 0.001). Step 3: genes common to all four cancers were selected for the core SOH signature (689 genes).

**Figure 2 cells-09-01761-f002:**
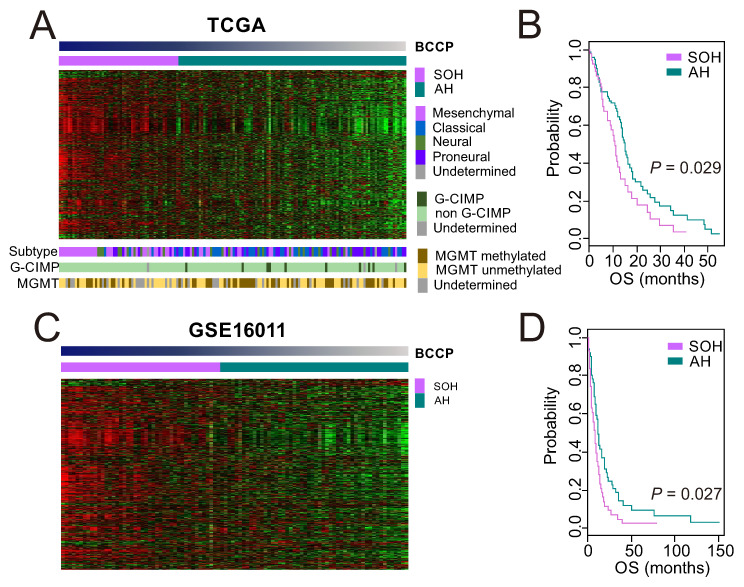
Association of the silence of Hippo pathway (SOH) signature with poor prognosis in patients with glioblastoma (GBM). (**A**) Heat map of gene expression patterns of GBM tumor samples from The Cancer Genome Atlas (TCGA; *n* = 154), which were stratified according to core SOH signature by the Bayesian covariate compound predictor (BCCP) classifier. Of 154 patient samples, 53 were placed in the SOH subgroup and 101 were placed in the active Hippo (AH) subgroup. The data are presented in matrix format in which rows represent individual genes and columns represent individual tumor samples. Samples are arranged according to SOH probability from BCCP. Mesenchymal subtypes were more common in the SOH subgroup, whereas proneural subtypes were more abundant in the AH subgroup (*p* < 0.001). The incidence of glioma CpG island methylation phenotype (G-CIMP) and promoter methylation of methylguanine methyl transferase (MGMT) are indicated. (**B**) Kaplan–Meier plots of overall survival (OS) of patients in the SOH and AH subgroups in TCGA. (**C**) Heatmap of gene expression data from GBM tumors in GSE16011 (44 SOH and 52 AH). (**D**) Kaplan–Meier plots of OS of patients in the SOH and AH subgroups in GSE16011. *p* values were obtained using the log-rank test.

**Figure 3 cells-09-01761-f003:**
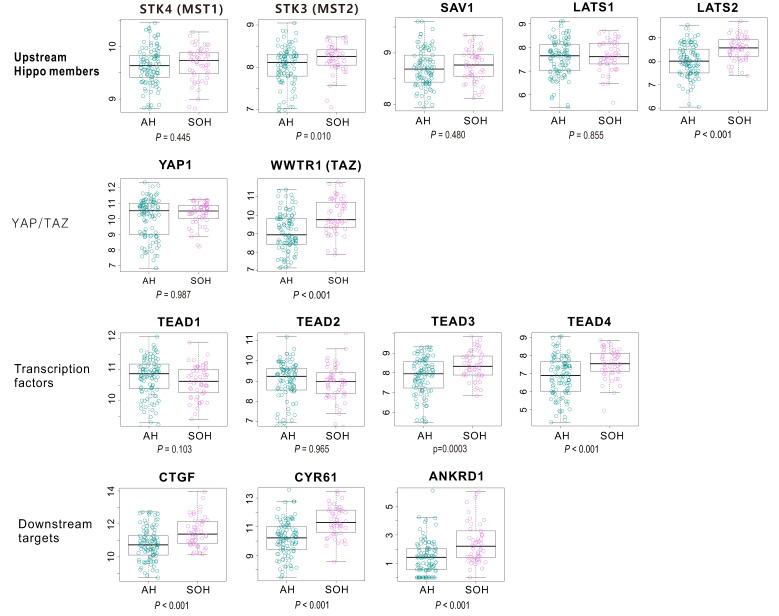
Differential gene expression levels of Hippo–YAP pathway-related genes in The Cancer Genome Atlas. The expression level of *TAZ* was elevated in the silence of Hippo (SOH) subgroup, although the gene expression level of YAP1 did not differ between the SOH and active Hippo (AH) subgroups. The SOH subgroup showed higher levels of *TEAD3* and *TEAD4* gene expression. Above all, the expression level of *CTGF* and *CYR61*, the most reliable and well-known downstream targets of *YAP1*, were markedly increased in the SOH subgroup. Differential gene expression of upstream members of Hippo pathway showed *MST2* and *LATS2* are upregulated in SOH subgroup revealing that these genes are not responsible for Hippo pathway inactivation.

**Figure 4 cells-09-01761-f004:**
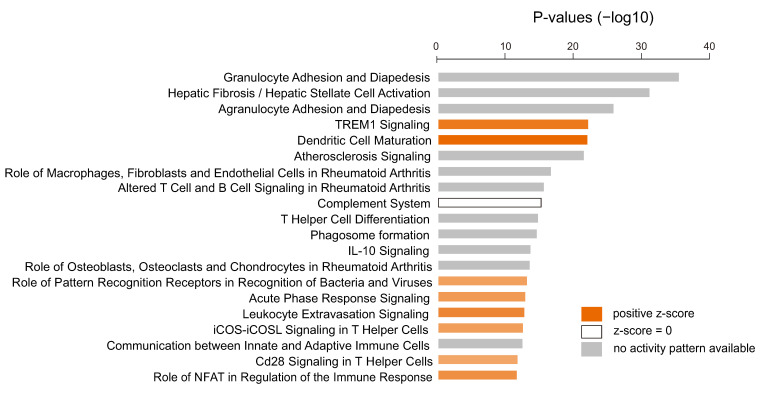
Pathway analysis based on 904 genes differentially expressed between the silence of Hippo (SOH) and active Hippo subgroups in The Cancer Genome Atlas. The top 20 canonical pathways are shown. Only pathways with significant positive z-score are marked with color. Various immune-associated pathways are significantly enriched in the SOH subgroup.

**Figure 5 cells-09-01761-f005:**
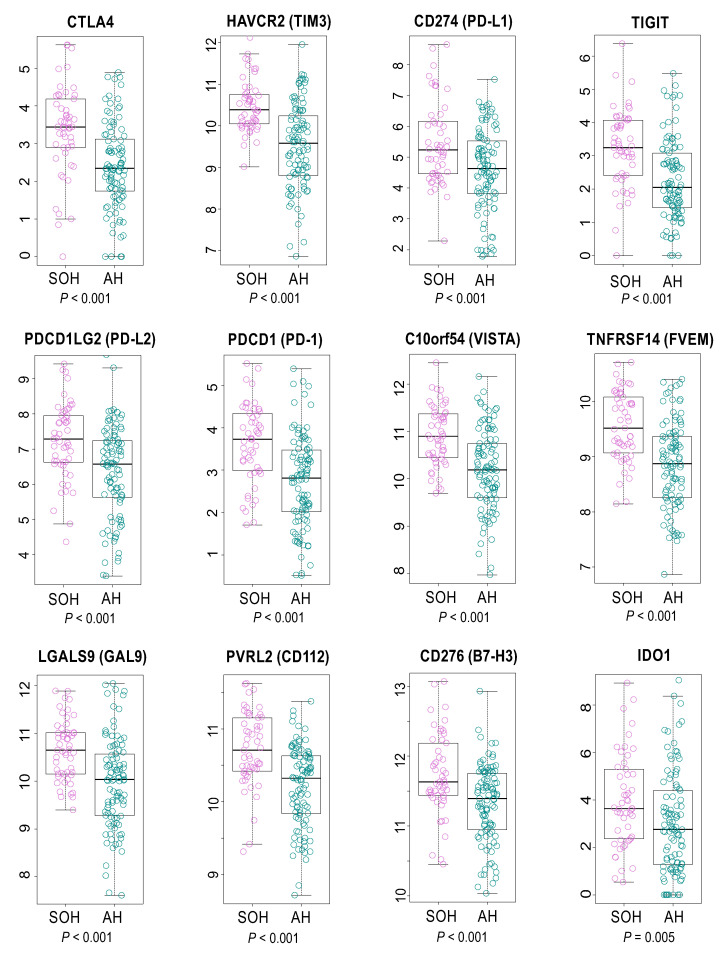
Gene expression levels of inhibitory checkpoint molecules in the silence of Hippo (SOH) and active Hippo (AH) subgroups in The Cancer Genome Atlas. Most inhibitory checkpoints were markedly upregulated in the SOH subgroup.

**Figure 6 cells-09-01761-f006:**
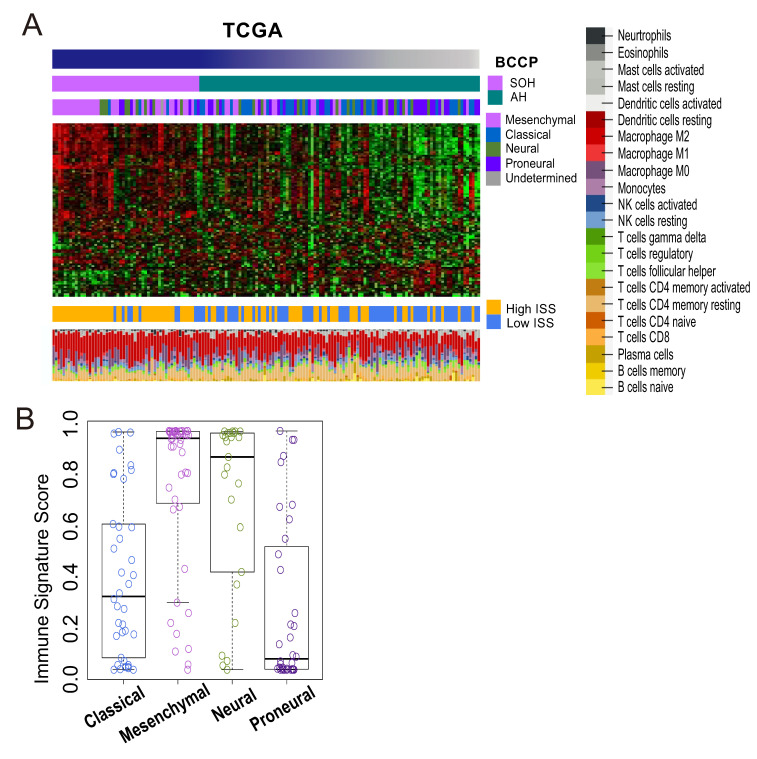
Immune signature of glioblastoma (GBM) samples from The Cancer Genome Atlas (TCGA). The immune signature score (ISS) was predicted for each of the 154 samples on the basis of a 105-gene immune signature generated from a comparative analysis of responders and nonresponders to immunotherapy. (**A**) In a heatmap showing the core immune signature of the 154 samples, high ISS was more common in the silence of Hippo (SOH) subgroup (*p* < 0.001). Leukocyte subset analysis (CIBERSORT) showed that the proportion of M2 macrophages in GBM tumor tissue was much higher in the SOH subgroup than in the active Hippo (AH) subgroup. BCCP, Bayesian covariate compound predictor. (**B**) ISS was apparently highest in the mesenchymal subtype, followed by the neural subtype, and it was lowest in the proneural subtype.

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
