# Peer review of "Silence of Hippo Pathway Associates with Pro-Tumoral Immunosuppression: Potential Therapeutic Target of Glioblastomas"

_cells, 2020, doi:10.3390/cells9081761_

Round 1
Reviewer 1 Report
In the present study, Kim et al. assessed the clinical relevance of the Hippo pathway in human glioblastoma (GBM). For this puprose, the authors generated a core gene expression signature from previously established four different gene signatures reflecting silence of the Hippo pathway (SOH). The authors found that the SOH signature was associated with poor prognosis in GBM. The expression levels of CTGF and CYR61 canonical targets of YAP1, were markedly increased in GBM with the SOH signature. In addition, the SOH signature was found to be associated with a high immune signature score and mesenchymal features. The authors conclude that inactivation of the Hippo pathway is significantly associated with poor prognosis in GBM, presumably due to protumoral immunosuppression.
The study is interesting and novel, providing intriguing insights into the molecular features of GBM progression and prognosis. However, the study was only based on association and not substantiated by in vitro findings, thus remaining mainly speculative. Furthermore, no confirmatory investigations on human specimens were conducted (immunohistochemistry, real-time or Western blot analysis), limiting the strength of the findings. Especially the immune-related data should be confirmed with an independent tool.
The study can be reassessed when these analyses are performed.
Reviewer 2 Report
The paper submitted by Dr. Eui Hyun Kim and Coworkers deals with the role played by YAL/TAZ and Hippo pathway on Glioblastoma development and progression.
The paper is overall very well organized, and very interesting the suggestions included in the conclusion section. First of all, the suggestion on the association btw activation of YAP/TAZ and poorer prognosis.
Very interesting the attempt to discover a trait common btw cancers developing in different tissues from different embryonal origin.
However, I have some suggestion:
Methods:
- I would performe an individual section on RPPA. I do not think that this method could be included in the Genome analysis
- Most relevant: on the “Molecular characteristics associated….” The authors state “Although expression of YAP1 was not significantly different btw the two subgroups….” It should be taken into account that the YAP function depends on its phosphorylation and on the site of the phosphorylation.
There are some typing mistakes:
Abs: line “in…in”
Introduction: last but one line: probably the word “modification “ (or something like) need to be added to “copy number amplification”.
Methods: § EMTS, 2nd line:
Reviewer 3 Report
Manuscript Eui Hyun Kim et al. "Silence of Hippo pathway associates with pro-tumoral immunosuppression: Potential therapeutic target of glioblastomas " is a thorough and reliable analysis of the role of the Hippo pathway in glioblastoma. The results obtained are interesting and interestingly presented and discussed. The studies described provide new biological insight into the poor prognosis associated with the SOH signature in GBM.
The manuscript is very well prepared in terms of content and is part of the research on the Hippo pathway disorders in cancer, conducted by the team and published in renowned scientific journals. The team's experience in conducting similar research translated into the substantive quality of the reviewed manuscript. I have a few small (mainly editorial) comments:
- 2 years ago, researchers published an abstract in Neuro-Oncology (Volume 20, Issue suppl_3, 1 September 2018, Page iii284, https://doi.org/10.1093/neuonc/noy139.261). The abstract has the same title and text as the abstract of the manuscript. In my opinion, it should be reworded to avoid self-plagiarism.
- Please differentiate between gene and protein names (eg. italics).
- There are many stylistic errors and formatting in the text (careless citations after the dot, no spaces, no justification of the text, capital letter in the middle of the sentence, repeated explanation of the same abbreviations (e.g. silence of the Hippo pathway (SOH) is used 9 times, active Hippo pathway (AH) is 6 times, The Cancer Genome Atlas (TCGA) - 4 times. The abbreviation should be explained once and used interchangeably with the full name).
- Figures are interesting and contain a lot of important information, but their quality is too low, which affects their readability (especially Figure 3 requires correction).
Reviewer 4 Report
Silence of Hippo pathway associated with pro-tumoral immunosuppression: Potential therapeutic target of glioblastomas.
Kim et al.
In this manuscript the authors built in previous studies where they have identified gene expression signatures for loss of canonical Hippo pathway signal in different cancers to study the role of this pathway in glioblastoma. The authors use publicly available data to identify a specific signature to this cancer type associated with loss of activity of the core kinases of the Hippo pathway and activation of YAP-dependent proliferative transcription program. Additionally, they use an algorithm to identify the immunological characteristics of glioblastomas in correlation with the hippo pathway status associated signatures. This analysis indicates that the tumors with a defective Hippo pathway show higher immunological components. Further bioinformatic analysis indicated that the TGF signalling may correlate with the macrophage markers.
While the description of a possible role of the Hippo pathway in glioblastoma may be of interest for the readership of Cells, the current manuscript is rather confusing and lacks reference to other articles where other groups have investigated the relationship between some of the members of the Hippo pathway and glioblastoma (at least 28 in a quick search in PubMed ie https://www.frontiersin.org/articles/10.3389/fgene.2019.00906/fullA Novel Prognostic Signature of Transcription Factors for the Prediction in Patients With GBM). It is current format there is a lack of explanation of the concepts behind the analysis that complicate the understanding of the work performed. Finally, there is no real experimental validation of the findings from the analysis which might not be necessary for the publication of the article but this make the claim that the finding may lead to a potential therapeutic intervention extremely speculative. There is a need to address several points.
Major points:
-The rational for their bioinformatics approach described in the section “generation of a core SOH signature and validation in GBM” is not clear to me. The tumors used are rather different and different driving mutations could have major effects. Why would these common genes be a genetic signature for Hippo pathway alterations? As far as I can see this could be related to other things irrespective of Hippo pathway silencing. How does the analysis show that this is related to SOH? And this relates to one major problem in the current manuscript which is that the authors do not explain what they consider silence of Hippo pathway (SOH) or activation of hippo (AH). Therefore, the reader not familiar with their previous work cannot understand what they do. When reading previous work from the group, the authors identified this SOH signature using MST1/2 and Salvador cells so I would expect that here they also consider that SOH would result in at least lower expression of these genes if this is referring to the same concept. However, Figure 3 shows that there is an increase of MST2 and LATS2 and no changes in MST1, SAV and LATS1 in the SOH glioblastomas. This seems to be a contradiction if one should expect that their analysis should identify as SOH tumors where those genes are lower. Is that correct or there is something I am missing? Thus, how can we be sure that the genetic signature identified is really related to the Hippo pathway. Since this is the basis of the whole study, explanation and validation of the finding is key to understand the work presented here and much work is necessary.
- Moreover, the fact that there is no difference in YAP expression but there is and increase in TAZ expression would indicate that this is related to TAZ alone so I do not understand how the can conclude as they state in the discussion “we showed that activation of YAP1/TAZ is associated with poor prognosis in GBM.” Again, there is a lack of explanation of what the authors define as activation of YAP/TAZ in this case. In the canonical view of the pathway this should be related to translocation to the nucleus upon inactivation of the core kinases. How would that happen here? Could it be possible that the signature is related to YAP/TAZ activity rather than the hippo pathway (similar to what the authors showed in CRC)?
-Additionally, the lack of change of expression of YAP in correlation with glioblastoma type seems to be at odds with a previous report where they used the TCGA data and they saw that changes in YAP expression correlates with globlastoma. Are the findings of these two reports contradictory? https://pubmed.ncbi.nlm.nih.gov/30009411/?from_term=hippo+glioblastoma&from_sort=date&from_page=2&from_pos=5
- The following sentence contributes to increase my confusion “When the gene expression levels of upstream members of Hippo pathway such as LATS1, LATS2, SAV1 were compared between SOH and AH subgroups, only MST2 and LATS2 were up-regulated in SOH subgroup, which revealed that they are not the leading cause of YAP activation”. If the core kinases that are supposed to be silence when the Hippo pathway is not active do not regulate whatever effect YAP is supposed to be mediated, is what the authors see here Hippo dependent at all? How do they explain this finding?
- The premise of the analysis is based on a canonical view of the Hippo pathway, that does not reflect a lot of the information that we have of the Hippo network. This is the case for the previous works from the group but in the case of neuronal cells it might be more of problem because what we know about the members of the pathway does not fit always with the canonical pathway. This is somehow acknowledged by the authors in the discussion when they address the limitations of the study (“predicted YAP1 activity in GBM might be underestimated due to lack of expression data reflecting neuronal specific YAP1 activity”). But in this case YAP isoforms and its regulation of the non-canonical TF p73 seems to be important in brain cells and might be more relevant that they estimate. (Up-regulation of Endogenous PML Induced by a Combination of Interferon-Beta and Temozolomide Enhances p73/YAP-mediated Apoptosis in Glioblastoma https://pubmed.ncbi.nlm.nih.gov/22542810/?from_single_result=yap+p73+glioblastoma and 10.1083/jcb.200509132. Some discussion about this might be necessary and it would be interesting to see if any of the differences could correlate with p73(p53) signatures.
-The statement “Although inactivation of the Hippo pathway is generally associated with copy number amplification of YAP1 rather than mutations in the Hippo pathway” is not really true. Silencing of core kinases and other proteins of the wider hippo network is much frequent than YAP/TAZ amplification. (10.3390/genes7060028).
-The discussion includes description of results that are not mentioned in the results section (namely supplementary data), and this should be rewritten and moved to the results sections.
Minor points
-References must be included to these studies in the introduction “In our previous studies, we established gene expression signatures reflecting silence of the Hippo pathway (SOH) in stomach, liver, ovarian, and colorectal cancers and demonstrated that SOH signatures were associated with poor prognosis in these four cancers”.
Round 2
Reviewer 1 Report
The authors have not addressed in any way my concerns. Thus, the data, although interesting, remain not cofirmed by independent methods. At least some immunohistochemistry on human cancer specimens (which can be easily purchased as tissue arrays) should have been performed.
Reviewer 3 Report
The authors have made the necessary corrections to the manuscript, which fully satisfy me. Thank you for all the comprehensive answers.
Congratulations on the results described and wish you quick publication.
Author Response
Dear reviewer
Thank you very much for kind words. We have done our best to address your comments, which greatly helped to improve our manuscript. We appreciate your thoughtful comment.
Reviewer 4 Report
The authors have addressed several of my comments and the manuscript is clearer.
As already mentioned in the previous review the lack of experimental validation detracts from the evaluation of the predictions. The authors acknowledge this limitation in the discussion. If this lack of experimental validation is in line with the journal editorial policy, I consider the manuscript may be of interest for the readership of cells.
There are some typos and grammatical mistakes in the review version that must be corrected.
Author Response
We again appreciate reviewer's thougtful and constructive comments as they greately helped to improve our manuscript. We also checked grammar and spellings again. Thank you very much.